# Antidiabetic Effect of Urolithin A in Cultured L6 Myotubes and Type 2 Diabetic Model KK-A$^y$/Ta Mice with Glucose Intolerance

Shinji Kondo [1],* , Shin-ichi Adachi [1], Wataru Komatsu [2], Fumiaki Yoshizawa [3,4] and Kazumi Yagasaki [1,4],*

1. Center for Bioscience Research and Education, Utsunomiya University, Utsunomiya 321-8505, Tochigi, Japan; adachi@tamateyama.ac.jp
2. Department of Public Health, Dokkyo Medical University School of Medicine, Mibu 321-0293, Tochigi, Japan; wkomatsu@dokkyomed.ac.jp
3. School of Agriculture, Utsunomiya University, Utsunomiya 321-8505, Tochigi, Japan; fumiaki@cc.utsunomiya-u.ac.jp
4. United Graduate School of Agricultural Science, Tokyo University of Agriculture and Technology, Fuchu 183-8509, Tokyo, Japan
* Correspondence: kondo.shinji.ga@u.tsukuba.ac.jp (S.K.); yagasaki@cc.tuat.ac.jp (K.Y.)

**Abstract:** Diabetes is caused by abnormal glucose metabolism, and muscle, the largest tissue in the human body, is largely involved. Urolithin A (UroA) is a major intestinal and microbial metabolite of ellagic acid and ellagitannins and is found in fruits such as strawberry and pomegranate. In this present study, we investigated the antidiabetic effects of UroA in L6 myotubes and in KK-A$^y$/Ta, a mouse model of type 2 diabetes (T2D). UroA treatment elevated the glucose uptake (GU) of L6 myotubes in the absence of insulin. This elevation in GU by UroA treatment was partially inhibited by the concurrent addition of LY294002, an inhibitor of phosphatidylinositol 3-kinase (PI3K) which activates Akt (PKB: protein kinase B) or Compound C, an inhibitor of 5′-adenosine monophosphate-activated protein kinase (AMPK). Moreover, UroA was found to activate both pathways of Akt and AMPK, and then to promote translocation of glucose transporter 4 (GLUT4) from the cytosol to the plasma membrane in L6 myotubes. Based on these in vitro findings, an intraperitoneal glucose tolerance test (IPGTT) was performed after the oral administration of UroA for 3 weeks to KK-A$^y$/Ta mice with glucose intolerance. UroA was demonstrated to alleviate glucose intolerance. These results suggest that UroA is a biofactor with antihyperglycemic effects in the T2D state.

**Keywords:** urolithin A; L6 myotube; PI3K; Akt; AMPK; GLUT4; glucose intolerance; IPGTT; KK-A$^y$/Ta mouse; type 2 diabetes





## 1. Introduction

There are three major types of diabetes: type 1 diabetes (T1D), type 2 diabetes (T2D), and diabetes in pregnancy (DIP). Among these, T2D accounts for approximately 90% of all diabetic patients [1], and this increasing trend is attributed to an aging, urbanized, and obesogenic environment [1]. The global prevalence of diabetes, especially T2D, is increasing, as reported in the 9th (2019) and 10th (2021) editions of the *Diabetes Atlas*. In 2019, an estimated 463 million people had diabetes [1]. In 2021, the number of people aged 20–79 years with diabetes worldwide was estimated to be 537 million, rising to 783 million by 2045 [2].

Diabetes is a mixture of metabolic disorders depicted by hyperglycemia resulting from flaws in insulin secretion, insulin action, or both. As chronic hyperglycemia of diabetes progresses, it leads to complications, of which diabetic retinopathy, diabetic nephropathy, and diabetic neuropathy are known as the three major complications of diabetes [3]. Thus, the most effective and simple way to prevent these complications seems to be restoring postprandial blood glucose levels to normal levels within several hours after each meal. Biofactors derived from edible sources such as phytochemicals and zoochemicals [4–7] seem

to be excellent candidates from the viewpoint of safety, as well as their antihyperglycemic effect, because they have a long history of ingestion.

Skeletal muscle is one of the largest tissues in our body and plays important roles in the metabolic regulations of key nutrients such as carbohydrates/glucose, lipids, and proteins/amino acids. In glucose metabolism, skeletal muscle is responsible for the majority (75%) of postprandial insulin-mediated glucose uptake and plays an important role in glucose homeostasis [8]. Isoquercetin, one of the flavonol glycosides, has been proven to stimulate glucose uptake (GU) in the isolated and incubated soleus muscle of rats under the condition of insulin absence by use of radioisotopes [9]. It is recognized that established cell lines are useful in terms of screening and mechanism analysis of many bioactive compounds. It is well known that L6 myoblasts established from rat skeletal muscle tissue spontaneously fuse and differentiate into multinucleated L6 myotubes under culture conditions [10], and the myotube was used to study the effects of biofactors on muscle function in vitro, without preparing muscle tissue from animals. Thus, we devised a simple, inexpensive, and rapid GU evaluation system under the condition of insulin absence, and no radioisotopes were used due to the application of the culture system [11]. We could find aspalathin, a component in rooibos tea, as the first natural compound that increased GU in this in vitro simple assay system [11], and its antihyperglycemic actions and their modes were studied in vivo by adopting KK-A$^y$/Ta, db/db, and ob/ob mice as the T2D model [11,12]. The assay of in vitro GU has been applied and is widely used by other research groups [13–15]. So far, we have screened approximately 2500 phytochemicals including aspalathin, while other phytochemicals, for example, resveratrol [16] and taxifolin [17], have been found to stimulate GU in cultured L6 myotubes, and to activate the pathways of phosphatidylinositol 3-kinase (PI3K)-Akt (PKB: protein kinase B) and/or 5′-adenosine monophosphate-activated protein kinase (AMPK), hence promoting the translocation of glucose transporter 4 (GLUT4) to the plasma membrane of L6 myotubes under insulin absence and showing an antihyperglycemic effect in T2D model db/db, ob/ob, and KK-A$^y$/Ta mice [12]. Among these compounds, enterolactone and equol are known as metabolites of lignans and daidzein, respectively, by intestinal microbiota [12]. These two metabolites were found to be effective at low doses in vivo [12].

Similarly, to equol and enterolactone, urolithins are produced by the gut microbiota from ellagic acid (EA) and ellagitannins (ETs) [18]. These metabolites, such as urolithin A (UroA), urolithin B (UroB), and urolithin C (UroC), are much better absorbed than their precursors and have been suggested to be responsible for the health-promotive effects attributed to EA and ETs that occur in food products such as pomegranates, berries, and nuts [19]. Safety assessment has indicated the remarkably safe profile of UroA in humans [20–22] and mice [23], and it has been reported to show various health promotive effects such as anti-inflammatory [24,25], hypouricemic [26], and improved actions on muscle strength and exercise performance in middle-aged adults [27]. Furthermore, UroA was reported to reduce fasting blood glucose levels in mice with insulin resistance induced by a high-fat/high-sucrose diet, while EA showed no such effect [28].

In the present study, we have tried to examine the effects of UroA on GU in cultured L6 myotubes in vitro, including signaling pathways related to GLUT4 translocation under insulin absence, and then examined the effects of UroA on glucose intolerance in T2D model KK-A$^y$/Ta mice in vivo.

## 2. Materials and Methods

### 2.1. Determination of Glucose Uptake in Cultured L6 Myotubes

Glucose uptake (GU) assay was conducted by the procedures described previously in L6 myotubes [11,16,17]. The rat-derived cell line of L6 myoblasts was purchased from American Type Culture Collection (ATCC®, CRL-1458, Manassas, VA, USA). The cells were maintained in Dulbecco's modified Eagle's medium (DMEM, 05919, Nissui Pharmaceutical Co., Tokyo, Japan) supplemented with L-glutamine (078-00525, FUJIFILM Wako Pure Chemical Corporation, Osaka, Japan), 0.06% sodium bicarbonate (191-01305, FUJIFILM

Wako Pure Chemical Corporation), 10% (*v/v*) fetal bovine serum (FBS, SH30070.03, Hy-Clone, Logan, UT, USA), and 1% penicillin–streptomycin mixed solution (09367-34, Nacalai Tesque, Inc., Kyoto, Japan; 100 U/mL penicillin and 100 μg/mL streptomycin as final doses in medium) in Nunc EasYDishes 100 mm type (150464, Thermo Fisher Scientific, Ltd., Waltham, MA, USA). The dishes were maintained under a condition of 95% humidified air/5% $CO_2$ at 37 °C, and myoblasts were successively applied to either GU assay in L6 myotubes or kept in liquid nitrogen as stocked myoblasts. Culture of myocytes, GU assay, and biochemical analyses were conducted essentially by the same procedures as described recently [17]. Briefly, L6 myoblasts at a density of $5 \times 10^4$ cells/well were seeded into a 24-well plate (MS-80240, Sumitomo Bakelite Co., Ltd., Tokyo, Japan) and proliferated to approximately 90% confluency in 10% FBS/DMEM (0.4 mL/well) for 3 days. Subsequently, for 7 days, the cells were differentiated to myotubes in 2% FBS/DMEM as a differentiation medium. The growth and differentiation medium were renewed every 2 days. For determination of GU, the myotubes were incubated in Krebs–Henseleit-HEPES buffer (KHH buffer) containing 3.3 mM $CaCl_2 \cdot 2H_2O$, 118 mM NaCl, 47.6 mM KCl, 7.3 mM $NaH_2PO_4 \cdot 2H_2O$, 1.2 mM $MgSO_4 \cdot 7H_2O$, and 25 mM $NaHCO_3$ (pH 7.4) supplemented with 2 mM sodium pyruvate, 10 mM HEPES, and 0.1% bovine serum albumin (BSA, fatty acid-free) (A8806-1G, Sigma-Aldrich Co., LLC., St. Louis, MO, USA) for 2 h. KHH buffer was filter-sterilized for use. Urolithin A (UroA, Z4919, Tokyo Chemical Industry Co., Ltd., Tokyo, Japan) was dissolved in dimethyl sulfoxide (DMSO, 046-21981, FUJIFILM Wako Pure Chemical Corporation) and finally diluted in KHH buffer containing 11 mM (198 mg/dL) glucose to final concentrations (UroA: 50–200 μM, DMSO: 0.1%) for treatment. Likewise, KHH buffer containing 11 mM glucose with 0 μM UroA and 0.1% DMSO was prepared as the control. The L6 myotubes were then incubated for another 4 h in KHH buffer containing glucose without or with each concentration of UroA. To conduct this research using inhibitors, 100 μM UroA contained in KHH buffer with glucose was co-treated without or with 10 μM Compound C (dorsomorphin) (040-33753, FUJIFILM Wako Pure Chemical Corporation) as an AMPK inhibitor, or 25 μM LY294002 (125-04863, FUJIFILM Wako Pure Chemical Corporation) as a PI3K inhibitor, for another 4 h after the 2 h incubation. Then, 1 nM human recombinant insulin (099-06473, FUJIFILM Wako Pure Chemical Corporation), an Akt activator, and 1 mM 5-aminoimidazole-4-carboxamide 1-β-D-ribofuranoside (AICAR) (015-22531, FUJIFILM Wako Pure Chemical Corporation), an AMPK activator, were employed as positive control agents, respectively. The differences in glucose contents of KHH buffer before and after incubation were determined by measuring the glucose concentrations using a Glucose CII-Test Kit (439-90901, FUJIFILM Wako Pure Chemical Corporation) with a plate reader SPARK 10M (TECAN, Männedorf, Switzerland) at 505 nm wavelength. The amounts of GU were calculated from the differences in glucose concentrations before and after treatment. Three similar experiments were conducted using the cells with different numbers of passages to confirm the results' reproducibility.

### 2.2. Extraction of Total Protein from Cultured L6 Myotubes

Protein extraction from L6 myotubes was conducted according to the procedure described recently [17]. L6 myoblasts at a density of $5 \times 10^5$ cells/dish were seeded into a 60 mm dish (150462, Thermo Fisher Scientific, Ltd.) and proliferated to approximately 90% confluency in 10% FBS/DMEM (3 mL/dish) for 3 days. After 24 h, the growth medium was replaced with 2% FBS/DMEM as a differentiation medium and then the myoblasts were differentiated into myotubes for 1 week. The fresh medium was replaced with a new one every 2 days. After incubation in KHH buffer for 2 h, L6 myotubes were incubated in 11 mM (198 mg/dL) glucose-contained KHH buffer without or with 100 μM UroA (0.1% DMSO) for another 15–60 min. Likewise, the control (0 μM UroA) was prepared with KHH buffer containing 11 mM glucose and 0.1% DMSO alone. For protein extraction, the myotubes were harvested from the dishes into RIPA buffer (containing a protease inhibitor cocktail, 08714-04, Nacalai Tesque) with 1% (*v/v*) phosphatase inhibitor cocktail (07575-51, Nacalai Tesque). The suspended cells were sonicated for 5 s on ice to produce lysates.

After centrifuging (10,000× *g*, 10 min, 4 °C), the supernatant was transferred into 1.5 mL tubes as an extracted protein. The extracts were mixed with 4% SDS at a ratio of 1:1 and then incubated (3 min, 100 °C). The incubated extracts were stored at −80 °C until used as samples to be run by electrophoresis in Western blotting.

### 2.3. Extraction of Total Protein from Plasma Membrane in Cultured L6 Myotubes

The fractions of the plasma membrane and the post-plasma membrane were harvested according to the procedures described by Nishiumi and Ashida [29], with minor modifications. For extraction of the cell membranes, the L6 myotubes were treated with 100 µM UroA for 60 min or positive controls with 1 nM insulin and 1 mM AICAR for 30 min. The myotubes were collected with buffer A containing 1% protease inhibitor cocktail (25955-24, Nacalai Tesque), 1% phosphatase inhibitor cocktail, 0.5 mM dithiothreitol (DTT, 048-29224, FUJIFILM Wako Pure Chemical Corporation), 50 mM Tris-HCl (pH 8.0), and 0.1% (*v/v*) IGEPAL CA-630 (198596, MP Biomedicals, Santa Ana, CA, USA). The collected myotubes were French-pressed at least 20 times using a syringe with a 27G needle (NN-2719S, Terumo Corporation, Tokyo, Japan). The homogenates were centrifuged (1000× *g*, 10 min, 4 °C) to separate the supernatant containing post-plasma membrane and the precipitate containing plasma membrane. The supernatants were put on ice for 1 h and then centrifuged (16,000× *g*, 20 min, 4 °C). After that, the supernatants were harvested as a fraction of the post-plasma membrane. On the other hand, the precipitates were suspended in IGEPAL CA-630-free buffer A for washing and centrifuged (1000× *g*, 10 min, 4 °C) and the washing was repeated. The precipitates were redissolved in buffer A containing 1.0% (*v/v*) IGEPAL CA-630 and French-pressed at least 20 times using a syringe with a 27G needle. The suspensions were put on ice for 1 h and centrifuged (16,000× *g*, 20 min, 4 °C). The obtained supernatants were harvested as a fraction of the plasma membrane [16]. The fractions of post-plasma membrane and plasma membrane were mixed with 4% SDS at a ratio of 1:1 and then incubated overnight at 4 °C to suppress aggregation.

### 2.4. Expression Analysis of Target Protein (Western Blotting)

To electrophorese the proteins, the concentrations of protein in each extract were quantified using a Pierce BCA Protein Assay Kit (23225, Thermo Fisher Scientific, Ltd.) and the plate reader at 562 nm wavelength. The proteins were loaded onto 10% polyacrylamide gels at an equal amount of 15 µg/lane and separated through electrophoresis at a voltage of 100 V. The separated proteins in the gels were transferred to Immuno-Blot® PVDF membranes (1620177, BIO-RAD, Hercules, CA, USA) at a voltage of 35 V for 5 h. The transferred membranes were blocked with 0.1% (*v/v*) Tween 20 contained 5% (*w/v*) BSA/Tris-buffered saline (BSA/TBST) at room temperature (1 h). After the blocking, the proteins on the membranes were reacted with the following primary antibodies: AMPK rabbit antibody (2532, Cell Signaling Technology, Beverly, MA, USA), p-AMPK (Thr172) rabbit antibody (2531, Cell Signaling Technology), Akt rabbit antibody (9272, Cell Signaling Technology), p-Akt (Ser473) rabbit antibody (9271, Cell Signaling Technology), GLUT4 (1F8) mouse antibody (2213, Cell Signaling Technology), and Na/K-ATPase rabbit antibody (3010, Cell Signaling Technology). The membranes reacting to the primary antibodies were incubated for 16 h at 4 °C with gentle shaking and then washed three times repeatedly with TBST. After the washing, the membranes were subsequently reacted with the following secondary antibodies: anti-mouse IgG HRP-linked whole antibody from sheep (NA931V, GE Healthcare UK Ltd., Buckinghamshire, UK) and anti-rabbit IgG HRP-linked whole antibody from donkey (NA934V, GE Healthcare UK Ltd.). The membranes reacting to the secondary antibodies were incubated for 1 h at room temperature with gentle shaking. After washing the membranes three times with TBST, the detection of the targeted proteins on membranes was enhanced by Amersham ECL Western Blotting Detection Reagent (RPN2106, GE Healthcare UK Ltd.) and detected using ChemiDoc XRS Plus with Image Lab Software ver. 3.0 for Windows (1708265J1NPC, BIO-RAD). These experimental methods followed the previous report [16], with appropriate modifications [17].

### 2.5. Intraperitoneal Glucose Tolerance Test in KK-A$^y$/Ta Mice

To determine the effect of UroA on an intraperitoneal glucose tolerance test (IPGTT), KK-A$^y$/TaJcl mice (KK-A$^y$/Ta, male, 4 weeks old, 16 mice) were adopted as model animals of T2D and C57BL/6JJcl (C57BL/6J, male, 4 weeks old, 7 mice) as non-diabetic normal animals. KK-A$^y$/Ta mice and C57BL/6J mice were obtained from CLEA Japan Inc. (Tokyo, Japan). All mice were kept in an individual cage in an air-conditioned room at a temperature of 22 °C, a relative humidity of 60%, and an 8:00–20:00 light/20:00–8:00 dark cycle. The mice were maintained on a standard mouse chow CRF-1 pellet diet (Oriental Yeast Co., Tokyo, Japan) and water ad libitum for the whole experimental period. After acclimating animals to the experimental environment for 1 week, the KK-A$^y$/Ta mice (5 weeks old) were divided into the following two groups with similar average body weights and blood glucose levels: diabetic control group (Control, 10 mice) and the oral administration group of UroA (100 mg/kg body weight/day) (UroA, 6 mice). The blood glucose levels were measured as described below. UroA was suspended in 0.3% (*w/v*) carboxymethyl cellulose sodium salt (CMC-Na; 039-01335, FUJIFILM Wako Pure Chemical Corporation) solution for oral administration. The UroA-suspended solution was orally administered once a day in the morning for 21 days at a dose of 3 mg/0.3 mL/30 g body weight (bwt)/day (100 mg/kg bwt/day) using a syringe with a sonde. The non-diabetic normal group (Normal, 7 mice of C57BL/6J strain) and diabetic control group (Control, 10 mice of KK-A$^y$/Ta strain) were orally administered 0.3% (*w/v*) CMC-Na solution alone for 21 days. Then, IPGTT was conducted as described previously [11], with slight modifications. The initial body weight and diet weight of all mice were measured immediately before the start of oral administration. On the day before the IPGTT, the final body weight and diet weight were measured at 20:00 on the 20th day of oral administration, after which the mice were deprived of their diets for fasting but were allowed free access to water. The food intake was determined by the difference in diet weight before and after oral administration until the start of fasting (20th day). After fasting for 15 h, UroA (100 mg/kg body weight) was orally administered to mice of the UroA group as usual on the 21st day (final oral administration). Likewise, 0.3% CMC-Na solution as a vehicle was orally administered to mice in the Normal and Control groups. The first blood samples (0 min) were collected from the tail vein of the mice 1 h after the final oral administration. Immediately after the first blood collection, the mice were injected intraperitoneally with glucose (0.2 g/mL/100 g body weight; 2 g/kg body weight) dissolved in saline. Blood samples were collected successively from the tail vein of each animal at precise time intervals (30, 60, 90, and 120 min) after the glucose injection. To evaluate the blood glucose levels, the collected blood (5 µL) was burst in 20 µL distilled water in a 1.5 mL tube on ice, then aqueous solution (25 µL) of 20% (*w/v*) trichloroacetic acid (T9159, Sigma-Aldrich Co., LLC.) was added to the tube and placed on ice. The mixtures at a volume of 50 µL were centrifuged (12,000× *g*, 5 min, 4 °C). The supernatant was collected for quantification of the blood glucose levels using the Glucose CII-Test Kit and the microplate reader at 505 nm wavelength. An area under the curve (AUC) of the blood glucose levels in the IPGTT was calculated by using the Prism 8 program ver. 8.3.4 (686) for Windows (GraphPad Software, San Diego, CA, USA). The animal experiment was conducted following the guidelines for the Animal Experiments of Utsunomiya University Animal Research Committee and was approved by this committee (Ethics Approval Number: A15-0017).

### 2.6. Statistical Analysis

All data are expressed as the mean values ± SEM. Data on GU and the phosphorylated protein (p-Akt/Akt and p-AMPK/AMPK) in L6 myotubes were analyzed by one-way ANOVA and Tukey–Kramer's multiple comparisons test as a post hoc test. Data on the GLUT4 translocation in L6 myotubes were analyzed by two-tailed Student's *t*-test. The treatments of insulin and AICAR (*n* = 1 for each) as positive controls in Western blotting were excluded from the statistical analyses. Data on the time-dependent changes in blood glucose levels and their AUC in the IPGTT were analyzed by two-way ANOVA and one-

way ANOVA, respectively, and Dunnett's multiple comparisons test as a post hoc test for both data. * $p < 0.05$ and ** $p < 0.01$ were considered statistically significant. These statistical analyses were conducted using the Prism 6 program for Mac OS X.

## 3. Results

### 3.1. Effect of UroA on Glucose Uptake in Cultured L6 Myotubes

We first evaluated the effect of UroA (Figure 1A) on GU of L6 myotubes under insulin absence (Figure 1B). UroA dose-dependently increased GU, and its effect was significant at concentrations of 100 and 200 μM. However, there was no significant change in GU between the 100 μM and 200 μM treatments. Thus, subsequent experiments were conducted at 100 μM UroA. We next examined the effects of inhibitors of PI3K and AMPK pathways on UroA-stimulated GU. LY294002 (25 μM) as a PI3K inhibitor and Compound C (10 μM) as an AMPK inhibitor partially but significantly canceled the UroA-induced promotion of GU (Figure 1C,D). Treatment with inhibitors alone, such as LY294002 (Figure 1C) or Compound C (Figure 1D), did not affect the basal GU. These results strongly suggest that UroA activates both PI3K/Akt and AMPK signaling pathways under the condition of insulin absence. However, either LY294002 alone (Figure 1C) or Compound C alone (Figure 1D) did not completely suppress the UroA-induced promotion in GU to the basal levels for non-treatments. In other words, GU was still significantly higher when UroA and each inhibitor were treated simultaneously, suggesting that other signaling pathways may be involved in the UroA action on GU in these cultured cells.

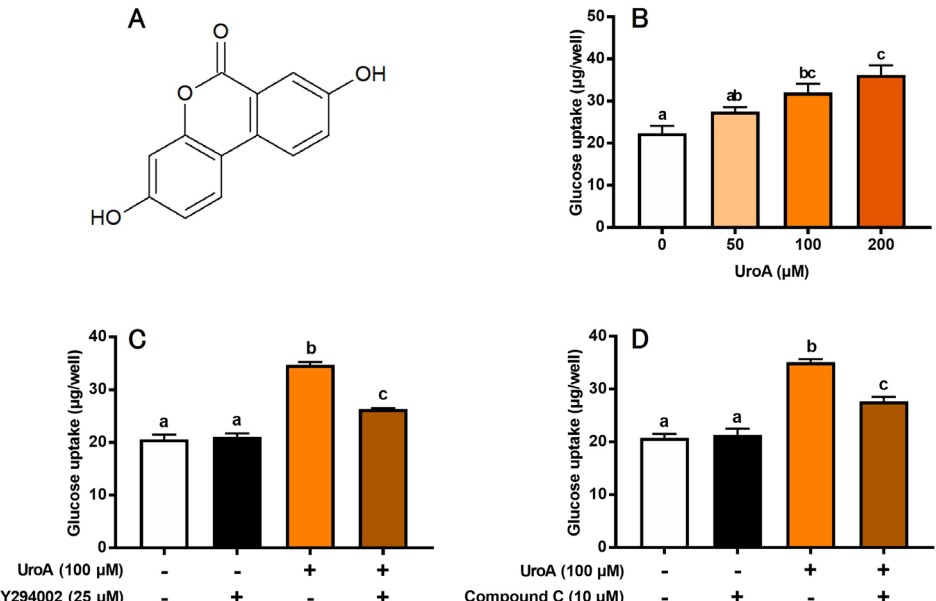

**Figure 1.** Structure of urolithin A (**A**). Glucose uptake on the dose-dependent effect of urolithin A (**B**), the effect of LY294002 (PI3K inhibitor) (**C**), and the effect of Compound C (AMPK inhibitor) (**D**) in L6 myotubes. Each value represents the mean ± SEM for 6 wells of a 24-well plate. Values not sharing a common letter are significantly different at $p < 0.05$ by the Tukey–Kramer multiple comparisons test.

### 3.2. Effect of UroA on Phosphorylation of Akt and AMPK in Cultured L6 Myotubes

To further analyze the mechanism of GU in L6 myotubes treated with UroA and each inhibitor, we confirmed phosphorylation of both Akt (p-Akt) and AMPK (p-AMPK) in UroA-treated L6 myotubes by Western blot analyses (Figure 2). L6 myotubes were treated with 100 μM UroA for 0–60 min or activators of Akt and AMPK (1 nM insulin as a PI3/Akt activator and 1 mM AICAR as an AMPK activator) for 30 min. The ratios of the phosphorylated protein levels (p-Akt and p-AMPK) were shown by normalizing the total protein levels of Akt and AMPK, respectively (p-Akt/Akt and p-AMPK/AMPK). The

phosphorylation ratios of Akt (p-Akt) and AMPK (p-AMPK) started to significantly increase 30 min after UroA treatment and thereafter further increased up to 60 min (Figure 2A,B).

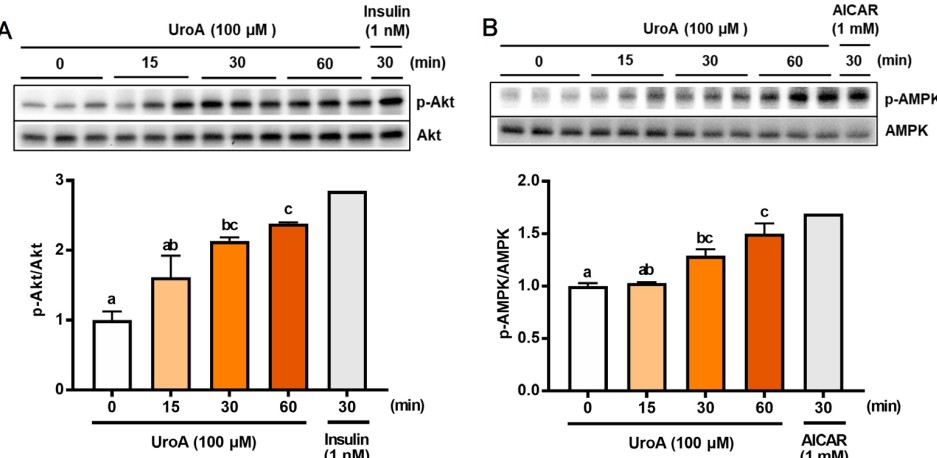

**Figure 2.** Effect of 100 μM urolithin A (15–60 min) on phosphorylation of Akt (**A**) and AMPK (**B**) in L6 myotubes. Each value represents the mean ± SEM for 3 dishes (60 mm diameter). Values not sharing a common letter are significantly different at $p < 0.05$ by the Tukey–Kramer multiple comparisons test.

### 3.3. Effect of UroA on GLUT4 Translocation to the Plasma Membrane in L6 Myotubes

To proceed to the investigation of mechanisms on UroA actions using T2D model KK-A$^y$/Ta mice, we finally examined the effect of UroA on GLUT4 translocation to the plasma membrane (PM) from the cytosol as post-plasma membrane (Post PM), induced by the phosphorylation of Akt and AMPK, in the L6 myotubes by Western blot analysis. L6 myotubes were treated with 100 μM UroA for 60 min as the highest activated time of Akt and AMPK, or positive controls (1 nM insulin and 1 mM AICAR) were treated for 30 min. The ratios of the GLUT4 protein levels in PM were shown by normalizing the Na$^+$/K$^+$-ATPase protein levels (GLUT4/Na$^+$/K$^+$-ATPase). In the PM fraction, as shown in Figure 3, UroA treatment significantly increased the GLUT4/Na$^+$/K$^+$-ATPase ratio higher than the treatment in the absence of UroA. The results indicated that the upregulation of GLUT4 translocation to PM by UroA treatment is considered to promote GU in L6 myotubes.

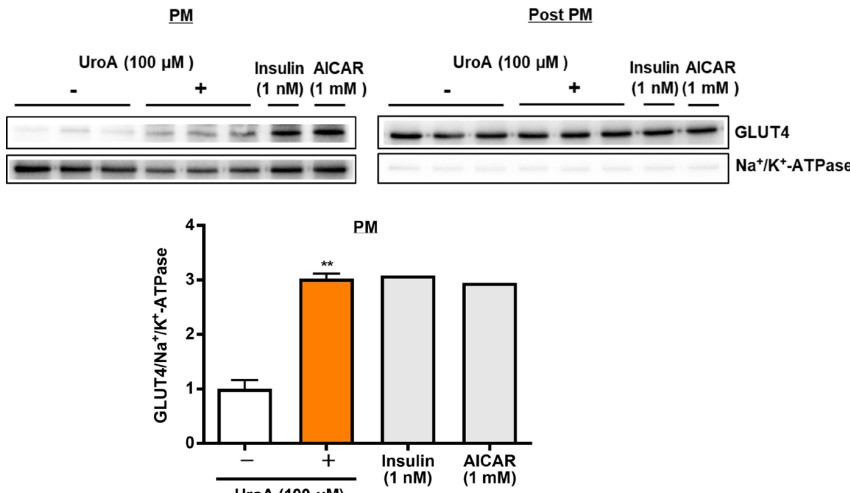

**Figure 3.** Effect of urolithin A on GLUT4 translocation to the plasma membrane from the cytosol (post-plasma membrane) in L6 myotubes. Each value represents the mean ± SEM for 3 dishes (60 mm diameter). Insulin and AICAR treatments were prepared as positive controls at n = 1 and were not considered in the statistical analyses. ** Statistically significant from no UroA treatment at ** $p < 0.01$ by two-tailed Student's *t*-test.

### 3.4. Effect of UroA on Intraperitoneal Glucose Tolerance Test in KK-A$^y$/Ta Mice

Based on the results of GU promotion by UroA and related mechanism analysis in L6 myotubes, we tried to examine the effect of UroA under a condition in T2D using KK-A$^y$/Ta mice. Initial body weights, body weight gains, and food intakes for the diabetic control mice (Control group) were significantly higher than for the non-diabetic normal mice (Normal group) throughout the 20 days from the start of oral administration to the start of fasting, despite the mice being the same age and having the same diet (Table 1), whereas there were no changes in body weight gain and food intake between UroA-treated diabetic mice (UroA group) and the Control group. In the IPGTT, as shown in Figure 4A, blood glucose levels in the Normal group rapidly peaked at 30 min after the intraperitoneal injection of glucose, and after that, blood glucose levels were gradually reduced up to the end (120 min). On the other hand, the levels of the Control group increased until 60 min and then remained high until the end. The oral administration of UroA significantly suppressed the rises in blood glucose levels at 90 and 120 min after the glucose injection. The blood glucose levels at 60 min for the UroA group were non-significantly suppressed compared to the Control group ($p$ = 0.12). In the AUC of IPGTT, the AUC in the Control group significantly increased compared with that of the Normal group, while UroA administration was exerted to suppress the increase of AUC in the Control group (Figure 4B).

**Table 1.** Effects of urolithin A on body weight (initial and final states), weight gain, and food intake for 20 days in C57BL/6J mice as non-diabetic normal model and KK-A$^y$/Ta mice as T2D model.

| Measurement | Normal | Control | UroA |
|---|---|---|---|
| Initial body weight (g) | 19.9 ± 0.3 ** | 28.8 ± 0.2 | 28.3 ± 0.7 |
| Final body weight (g) | 22.3 ± 0.4 ** | 36.9 ± 0.4 | 36.1 ± 1.0 |
| Weight gain (g/3 weeks) | 2.4 ± 0.4 ** | 8.0 ± 0.4 | 8.0 ± 0.5 |
| Food intake (g/3 weeks) | 60.8 ± 1.4 ** | 111.3 ± 2.6 | 110.8 ± 4.2 |

Non-diabetic C57BL/6J mice (Normal) and diabetic model KK-A$^y$/Ta mice (Control) received 0.3% sodium carboxymethyl cellulose (CMC) solution, while urolithin A (UroA) suspended in CMC was orally administered to KK-A$^y$/Ta mice once a day for 21 days at a dose of 100 mg/kg bwt/day. The initial body weight and diet weight of all mice were measured before the start of oral administration. Final body weight and diet weight were measured before the start of fasting on the 20th day of oral administration, the day before IPGTT. The food intake was determined by the difference in diet weight before and after oral administration until the start of fasting (20th day). Each value represents the mean ± SEM for 7 (Normal), 10 (Control), and 6 (UroA) mice. ** Significantly different from the Control group at ** $p$ < 0.01 by Dunnett's multiple comparisons test.

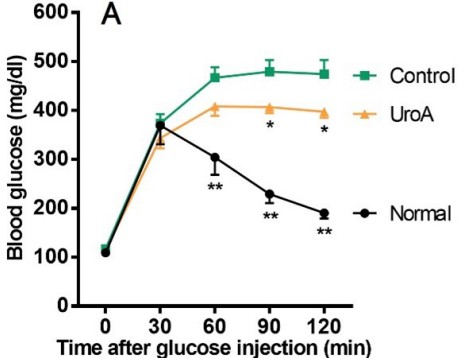 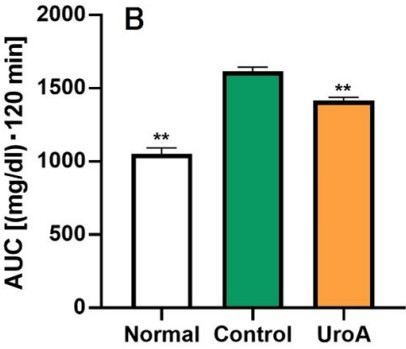

**Figure 4.** Effect of urolithin A on intraperitoneal glucose tolerance test (**A**) and area under the curve (**B**) in type 2 diabetic model KK-A$^y$/Ta mice. (**A**) IPGTT was conducted using Normal, Control, and UroA groups after oral administration of vehicle or UroA for 3 weeks. (**B**) AUC was calculated from (**A**). Each value represents the mean ± SEM for 7 (Normal), 10 (Control), and 6 (UroA) mice. Data on the time-dependent changes in blood glucose levels and their AUC in the IPGTT were analyzed by two-way ANOVA and one-way ANOVA, respectively, and Dunnett's multiple comparisons test as a post hoc test in both data. * $p$ < 0.05 and ** $p$ < 0.01 were considered to be statistically significant compared to the Control group.

## 4. Discussion

In the present study, we found that UroA treatment promotes GU via GLUT4 by activating PI3K/Akt and AMPK pathways in L6 myotubes. Furthermore, we performed IPGTT using KK-A$^y$/Ta, a mouse model of T2D, and confirmed that oral administration of UroA suppresses the increase in the level of blood glucose (Figure 5).

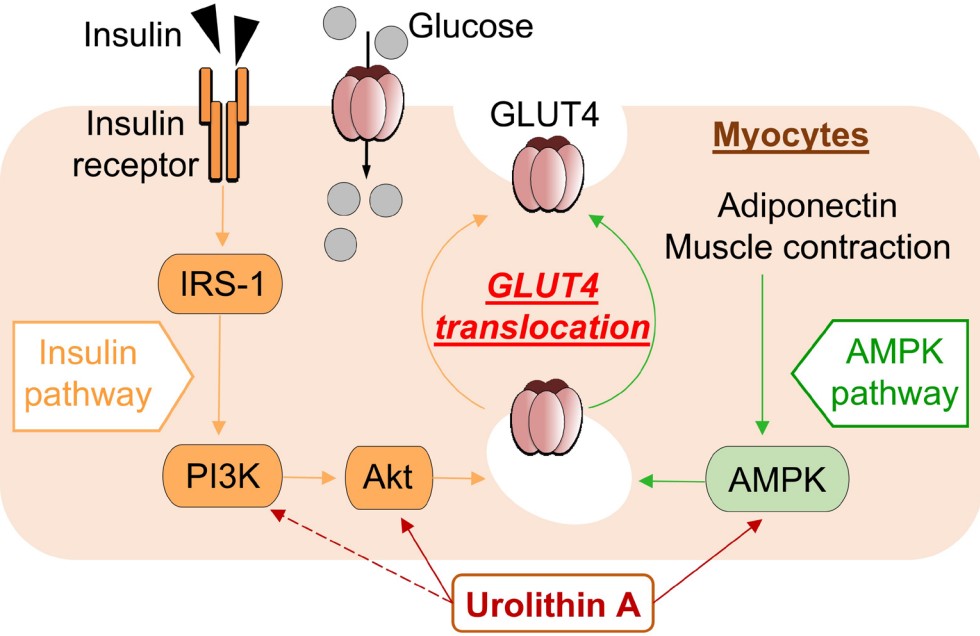

**Figure 5.** Schematic diagram showing pathway mechanisms involved in glucose uptake into myocytes from extracellular glucose pools.

The development of metabolic diseases such as hyperglycemia is well known to be associated with insulin resistance in skeletal muscle. Since it has been reported that the treatment of isolated rat muscle with the AMPK activator AICAR increases GU via GLUT4 in an insulin-independent manner, the regulation of AMPK as well as insulin contributes to diabetes control [30,31]. In recent years, several studies have reported the discovery of natural bioactive compounds to improve insulin resistance associated with T2D [32–34]. In the present study, UroA significantly enhanced GU by L6 myotubes in the insulin absence and oral administration of UroA to KK-A$^y$/Ta mice, a mouse model of T2D exhibiting insulin resistance [35,36], and suppressed blood glucose elevation in IPGTT.

Physiologically, insulin and adiponectin activate the PI3K/Akt and AMPK pathways, respectively, resulting in GU via GLUT4 membrane translocation [37,38]. In T2D, insulin resistance is increased, resulting in increased blood insulin levels, and blood adiponectin levels are decreased [39,40]. Both responses inhibit the PI3K/Akt and AMPK activations and reduce GLUT4-mediated GU in the diabetes condition. The KK-A$^y$/Ta mice were reported to increase levels of plasma insulin and to decrease levels of plasma adiponectin compared to healthy mice, suggesting that reversal of these effects may contribute to diabetes prevention [17]. The present study suggests that UroA activates PI3K/Akt and AMPK pathways and GLUT4-mediated GU in L6 myotubes in the absence of insulin and adiponectin. On the other hand, in the in vivo IPGTT study, oral administration of UroA to KK-A$^y$/Ta mice suppressed the elevated blood glucose levels after intraperitoneal administration of sugar solution.

T2D is prone to hyperlipidemia, and free fatty acids in the blood have been reported to induce insulin resistance [41]. One of the reasons for this is that free fatty acids inhibit tyrosine phosphorylation in insulin receptor substrate-1 (IRS-1) and impair activation of the PI3K/Akt pathway. It has been reported that UroA administration suppresses elevated blood insulin levels in high-fat diet-fed mice [42]. This suggests that UroA alleviates insulin

resistance in muscle and induces the activation of the PI3K/Akt pathway, which results in GU increase via GLUT4 membrane translocation. Furthermore, it was reported that UroA administration increased the level of blood adiponectin in high-fat diet-fed mice [43]. This suggests that an increase in the level of blood adiponectin induced the activation of the AMPK pathway, which may have caused GU via GLUT4 membrane translocation. It was reported that the administration of UroA to high-fat diet-fed mice increased energy expenditure and prevented hyperlipidemia by enhancing thermogenesis in brown adipose tissue and inducing the browning of white adipose tissue, decreasing insulin resistance [42].

In the present study, UroA was shown to increase GU in cultured L6 myotubes in the absence of insulin in a dose-dependent manner via both the PI3K/Akt and AMPK signaling pathways, and similar effects have been reported with naturally occurring bioactive components such as resveratrol [16]. The activation of each pathway independently promoted the translocation of GLUT4 to the plasma membrane from the cytosol (Figure 5). On the other hand, unlike UroA, Cyanidin 3-*O*-glucoside was found to promote GU of L6 myotubes via the PI3K/Akt pathway only [44]. The difference in effects of those naturally occurring bioactive components may depend on the specificity of their chemical structures.

The interaction of glucose utilization by GLUT4 via the PI3K/Akt and AMPK pathways, as induced by UroA in the present study, with the regulation of reactive oxygen species (ROS)/inflammation was supported by studies using a bioactive compound and T2D model mice [45]. It is known that insulin resistance of skeletal muscle is induced in T2D via increased ROS and inflammatory cytokines (IL-6, TNF$\alpha$, etc.) [46,47]. Several natural bioactive compounds like resveratrol, with antioxidant and anti-inflammatory properties, have been reported to alleviate insulin resistance [48,49]. Since UroA has also been reported to suppress ROS production and LPS-induced inflammatory cytokine production in RAW264 macrophages [25], UroA may prevent increased skeletal muscle insulin resistance in type II diabetes. In addition, although Akt and AMPK activation work antagonistically with each other [50], it appears that the activation of both pathways via the antioxidant and anti-inflammatory effects of UroA is predominant over their agonistic actions.

Currently, no in vitro experiments have been performed in L6 myotubes under conditions that reproduce the T2D environment, i.e., with insulin resistance. In vitro experiments should be performed in the presence of insulin to treat L6 myotubes with UroA. In vivo studies should also examine the effects of UroA on the PI3K/Akt and AMPK pathways in the skeletal muscle of KK-A$^y$/Ta mice, similarly, to the in vitro studies conducted in this study. Regarding the intensity of UroA dose-dependent activity, in the present study, oral administration of 100 mg/kg bwt UroA suppressed the elevation of blood glucose levels in T2D model mice. In a study at higher dosages, oral administration of 200 mg/kg bwt UroA reduced the risk of developing Alzheimer's disease in a model mouse of T2D, which shows an epidemiological correlation with Alzheimer's disease [51]. Therefore, UroA doses higher than 100 mg/kg may be worth considering, as they may be even more effective in suppressing the elevation of blood glucose levels.

## 5. Conclusions

We found that one of the mechanisms of UroA action is the activation of both PI3K/Akt and AMPK, followed by the translocation of GLUT4 to the plasma membrane of muscle cells, the main site where GU takes place. We also found that UroA significantly suppressed the elevation of blood glucose levels in IPGTT using KK-A$^y$/Ta mice. These results suggest that UroA has a hyperglycemic inhibitory effect on the development of T2D, which may be mediated by a major mechanism involving the induction of GLUT4 membrane translocation via the activation of both PI3K/Akt and AMPK.

**Author Contributions:** The following individual efforts contributed to this study: Conceptualization, K.Y. and S.K.; Methodology, S.K., S.-i.A. and K.Y.; Investigation, S.K. and S.-i.A.; Data curation, S.K.; Writing—original draft preparation, S.K. and K.Y.; Writing—review and editing, S.K., S.-i.A., W.K., F.Y. and K.Y.; Software, S.K.; Supervision, K.Y.; Project administration, K.Y. and F.Y.; Funding acquisition, F.Y. and K.Y. All authors have read and agreed to the published version of the manuscript.

**Funding:** This study was funded in part by both the Japan Society for the Promotion of Science (JSPS KAKENHI grant number 15K07424 to K.Y.) and the Regional Innovation Strategy Support Program, MEXT, Japan.

**Institutional Review Board Statement:** The animal experiment conducted in the present study was performed under the guidelines for the Animal Experiments of Utsunomiya University Animal Research Committee. The committee approved the animal experiment (Ethics Approval Number: A15-0017).

**Informed Consent Statement:** Not applicable.

**Data Availability Statement:** All data used and/or analyzed in the present study are available from the corresponding authors upon reasonable request.

**Acknowledgments:** The authors would like to express their great appreciation to Keiichiro Numao, Madoka Ushio, Miko Tasaki, Reiko Suzuki, Ryuji Hayashi, and Takuya Namai for their excellent technical support in laboratory mouse husbandry and time-dependent blood sampling.

**Conflicts of Interest:** The authors declare no conflicts of interest.

## Abbreviations

| | |
|---|---|
| AICAR | 5-aminoimidazole-4-carboxamide 1-β-D-ribofuranoside |
| Akt | protein kinase B |
| AMPK | 5′-adenosine monophosphate-activated protein kinase |
| BSA | bovine serum albumin |
| DTT | dithiothreitol |
| DMEM | Dulbecco's modified Eagle's medium |
| FBS | fetal bovine serum |
| GLUT4 | glucose transporter 4 |
| GU | glucose uptake |
| IPGTT | intraperitoneal glucose tolerance test |
| KHH buffer | Krebs–Henseleit-HEPES buffer |
| PI3K | phosphatidylinositol 3-kinase |
| TBST | Tris-buffered saline with Tween 20 |
| T2D | type 2 diabetes |
| UroA | urolithin A |

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
