# Peer review of "Antidiabetic Effect of Urolithin A in Cultured L6 Myotubes and Type 2 Diabetic Model KK-Ay/Ta Mice with Glucose Intolerance"

_cimb, doi:10.3390/cimb46020068_

Round 1
Reviewer 1 Report
Comments and Suggestions for Authors
This paper explores the inhibition of blood glucose levels by Urolithin A in vivo and in vitro. A deeper exploration of the in vivo anti-diabetic pathway was conducted. The study demonstrated the great potential of Urolithin A for anti-diabetes. However, the effects of Urolithin A compared to insulin in diabetic mice are not clear, and the toxicity of Urolithin A is not clear. Therefore, additional experiments are needed, and the paper needs to be further revised. Here are some comments.
1. In the abstract, this paper writes about muscle and Urolithin A. Is there any connection between muscle and urolithin A? Please write the abstract carefully to ensure that it is logically accurate.
2. In line 265, Urolithin A (200 μM) increased the GU effect significantly, why was 100 μM UroA chosen for subsequent experiments? There was no detailed explanation in the text, please explain the additional reasons.
3. Please add experiments related to the toxicity of Urolithin A.
4. Regarding the fact that Urolithin A was not ideal for lowering blood glucose levels and controlling body weight in mice, would the use of 200 μM Urolithin A be more effective in controlling body weight in mice?
5. Please discuss in detail the antidiabetic activity of Urolithin A compared to other antidiabetic natural products.
6. Why was insulin not used as a control for comparison in the intraperitoneal glucose tolerance test of urolithin in KK-Ay/Ta mice, a model of type 2 diabetes? Please add a comparison between Urolithin A and insulin.
7. For references, please try to select citations from papers from recent years.
Comments on the Quality of English LanguageMinor editing of English language required
Author Response
Dear Reviewer 1
Thank you very much for providing important comments. We are thankful for the time and energy you expended.
Please see the file for our response to your comments.
Sincerely yours,
Shinji Kondo, Kazumi Yagasaki

Reviewer 2 Report
Comments and Suggestions for Authors
Dear authors,
I read the manuscript entitled “Antidiabetic Effect of Urolithin A in Cultured L6 Myotubes and Type 2 Diabetic Model KK-Ay/Ta mice with Glucose Intolerance”.
This study aims to analyze the effects of Urolithin A (UroA), a metabolite derived from the degradation of ellagic acid and ellagitannins, on L6 myotubes and type 2 diabetes model mice. The authors show that treatment with UroA stimulates glucose uptake in L6 myotubes and this effect is only partially reversed by pretreatment with LY294002, a PI3K inhibitor, or compound C, an AMPK inhibitor. Furthermore, the authors demonstrated that UroA treatment increases the phosphorylation levels and activation of both Akt and AMPK, which in turn promote the translocation of GLUT4 to the plasma membrane of myotubes. Finally, the authors demonstrated that pretreatment with UroA in model mice with type 2 diabetes (KK-Ay/TaJcl mice), leads to a significant reduction in glucose levels in the blood at 90 and 120 minutes after an intraperitoneal glucose tolerance test (IPGTT). Therefore, based on these evidences, the authors concluded that UroA shows antihyperglycemic properties, and that these are mediated by the activation of both PI3K/Akt and AMPK.
Overall, I consider this manuscript interesting, well organized and writen.
However, to better describe the mechanism of action of UroA, I believe that more attention should be paid to the mechanisms that promote the activation of Akt and AMPK, two important enzymes that exhibit antagonistic activity. Since both enzymes are activated by stress conditions and ROS, it might be interesting to evaluate whether UroA activity is mediated by increased ROS levels in both L6 myotubes and type 2 diabetes model mice.
Finally, the discussion section should be implemented taking into account the antagonistic activity of Akt and AMPK.
Author Response
Dear Reviewer 2
Thank you very much for providing important comments. We are thankful for the time and energy you expended.
Please see the file for our response to your comments.
Sincerely yours,
Shinji Kondo, Kazumi Yagasaki

Round 2
Reviewer 1 Report
Comments and Suggestions for Authors
-